# Complete Mitochondrial Genome for *Lucilia cuprina dorsalis* (Diptera: Calliphoridae) from the Northern Territory, Australia

**DOI:** 10.3390/genes15040506

**Published:** 2024-04-18

**Authors:** Shilpa Kapoor, Ying Ting Yang, Robyn N. Hall, Robin B. Gasser, Vernon M. Bowles, Trent Perry, Clare A. Anstead

**Affiliations:** 1Bio21 Molecular Science and Biotechnology Institute, The University of Melbourne, Parkville, VIC 3010, Australia; ytyang@unimelb.edu.au (Y.T.Y.); trentp@unimelb.edu.au (T.P.); 2Department of Veterinary Biosciences, Melbourne Veterinary School, Faculty of Science, The University of Melbourne, Parkville, VIC 3010, Australia; robinbg@unimelb.edu.au (R.B.G.); vmb@unimelb.edu.au (V.M.B.); 3CSIRO Health & Biosecurity, Acton, ACT 2601, Australia; robyn.hall@ausvet.com.au; 4Ausvet Pty Ltd., Fremantle, WA 6160, Australia

**Keywords:** Australian sheep blowfly, *Lucilia cuprina dorsalis*, flystrike, mitochondrial genome, phylogenetic analyses

## Abstract

The Australian sheep blowfly, *Lucilia cuprina dorsalis*, is a major sheep ectoparasite causing subcutaneous myiasis (flystrike), which can lead to reduced livestock productivity and, in severe instances, death of the affected animals. It is also a primary colonizer of carrion, an efficient pollinator, and used in maggot debridement therapy and forensic investigations. In this study, we report the complete mitochondrial (mt) genome of *L. c. dorsalis* from the Northern Territory (NT), Australia, where sheep are prohibited animals, unlike the rest of Australia. The mt genome is 15,943 bp in length, comprising 13 protein-coding genes (PCGs), two ribosomal RNAs (rRNAs), 22 transfer RNAs (tRNAs), and a non-coding control region. The gene order of the current mt genome is consistent with the previously published *L. cuprina* mt genomes. Nucleotide composition revealed an AT bias, accounting for 77.5% of total mt genome nucleotides. Phylogenetic analyses of 56 species/taxa of dipterans indicated that *L. c. dorsalis* and *L. sericata* are the closest among all sibling species of the genus *Lucilia*, which helps to explain species evolution within the family Luciliinae. This study provides the first complete mt genome sequence for *L. c. dorsalis* derived from the NT, Australia to facilitate species identification and the examination of the evolutionary history of these blowflies.

## 1. Introduction

The Australian sheep blowfly, *Lucilia cuprina dorsalis* (Robineau-Desvoidy, 1830), is one of the most important ectoparasites of sheep in Australia. Female *L. c. dorsalis* deposit their eggs on soiled wool, in wounds, or other decaying organic matter [1]. Subsequently, the eggs hatch into larvae, commonly referred to as maggots, which then feed on skin secretions, dermal tissues, and/or the blood of sheep [2]. The presence of this invasive species is responsible for ~90% of flystrike incidents in Australia and imposes a significant economic strain of ~$324 million each year on the wool industry in this country, due to the expenses associated with prevention and control measures [3]. Furthermore, flystrike gives rise to substantial animal welfare concerns for both sheep meat and wool producers, due to the direct effect of the parasite and due to some controversial methods of control (e.g., mulesing) [4]. This blowfly is also a primary colonizer of fresh carrion [5], an important species in maggot debridement therapy and forensic investigations [6], and has also been documented as an efficient pollinator of avocado [7].

Another *Lucilia* subspecies, *L. cuprina cuprina* (Wiedemann, 1830), exists in Australia but typically inhabits urban areas [8,9]. Distinguishing between the two subspecies, *Lucilia c. cuprina* and *L. c. dorsalis*, presents a considerable challenge, due to their striking morphological similarity, making microscopic differentiation difficult [10]. These two subspecies have been shown to interbreed, producing hybrids that share morphological features, which further complicates their identification [9]. The common green bottle fly, *Lucilia sericata* (Meigen), is also found in Australia and can hybridize with *L. cuprina dorsalis* [9,11,12]. The potential for hybridization between these *L. cuprina* subspecies, as well as between *L. c. dorsalis* and *L. sericata,* highlights a need to investigate the genetic differences between these blowflies that exhibit very different feeding preferences (e.g., live sheep and carrion).

Exploring the mitochondrial (mt) genomes of blowflies underpins genetic and systematic studies of dipterans, substantially enhancing the accuracy of the identification of species and detection of cryptic (hidden) species and hybrids [10,12] using molecular markers. By sequencing the complete mt genomes of *Lucilia* specimens from various geographical regions across Australia, valuable data can be gathered to create efficient and dependable molecular tools for accurate species identification and differentiation, helping to resolve phylogenetic relationships [8,9]. In this study, the complete mt genome of *L. c. dorsalis* originating from the Northern Territory (NT), Australia, was assembled, annotated, and compared with the mt genomes of a wide range of species/taxa of the order Diptera (GenBank) to study phylogenetic relationships amongst these flies. Blowflies from the NT were selected because *L. c. dorsalis* is almost exclusively parasitic in Australia; however, sheep are prohibited animals in the NT, due to the presence of the bluetongue virus and its potential impact on the livestock industry [13]. The identification and molecular characterization of this blowfly species across its entire Australian geographic distribution is an important step in understanding invasion events and migration patterns that may assist with the development of interventions targeted to species biology and behavior. 

## 2. Material and Methods

### 2.1. Sample Collection 

Between April 2017 and July 2019, samples of *Lucilia* were collected from the NT (AZRI site; 23°45′58.1″ S 133°52′46.0″ E), Australia using Envirosafe™ fly traps (Bunnings, Perth, Australia), as described previously [14]. Flies were stored frozen at −20 °C until being transferred into RNAlater (Thermo Fisher Scientific, Waltham, MA, USA) for shipment and storage at the School of BioSciences, The University of Melbourne, Australia. Adult flies were identified as *L. c. dorsalis* using morphological characters [15,16,17,18]. 

### 2.2. DNA Extraction, Library Construction, and Sequencing

Genomic DNA (gDNA) was isolated from the head of each of the 17 blowflies using a well-established method [8,19,20]. The quality of the DNA was evaluated visually through electrophoresis on a 1% agarose gel, and its quantity was measured using a Qubit Fluorometer (Invitrogen, Waltham, MA, USA). Subsequently, a composite DNA sample was prepared by combining equal amounts of DNA from all 17 individual flies. The DNA library was then prepared using the NEBNext^®^ Ultra™ II DNA Library Prep Kit (Ipswich, MA, USA), followed by paired-end sequencing using 2 × 150 cycles on the Illumina NovaSeq 6000 platform (San Diego, CA, USA).

### 2.3. Mitochondrial Genome Assembly and Annotation 

The consensus mt genome (BioProject ID PRJNA419080, GenBank accession number: PP297113) representing *L. c. dorsalis* from the NT, Australia was assembled. Adapters, contaminants, low-quality sequencing reads (Phred scores < 30), and reads shorter than 50 base pairs (bp) were eliminated using the Trimmomatic program v.0.39 [21]. After filtering, the quality of the reads was assessed using FastQC v.0.11.9 [22], and a de novo assembly was built with the program NOVOPlasty v.4.2 using the remaining high-quality sequencing reads [23]. The mt DNA sequence was annotated using the MITOS2 web server [24] using NCBI’s Invertebrate Mitochondrial translation Table 5. Additionally, the ARWEN v.1.2.3 software was used to identify transfer RNA (tRNA) genes [25]. Protein-coding genes (PCGs) were translated based on NCBI’s Invertebrate Mitochondrial translation in Table 5 and manually curated to ensure the functionality of each PCG-encoded protein. Visualization of the mitochondrial genome was performed using Geneious Prime v.2019.2.3 [26].

### 2.4. Genomic and Phylogenetic Analyses

Geneious Prime v.2019.2.3 [26] was used to determine the base composition and nucleotide frequencies, with the formulas (AT skew  =  (A −  T)/(A  +  T) and GC skew  =  (G − C)/(G  +  C)) used to calculate nucleotide composition and skewness, respectively [27]. Bowtie2 v2.4.5 [28] was used to align the *L. c. dorsalis* (NT, Australia) FASTQ reads with the *L. c. dorsalis* mt reference genome (GenBank accession number: MW255536; VIC, Australia). The “Find Variations/SNPs” tool within the Annotate and Predict function of the Geneious Prime v.2019.2.3 software [26] was used to extract the nucleotide polymorphisms, with minimum coverage and minimum variant allelic frequency thresholds set to 5 and 0.25, respectively.

The phylogenetic relationships between *L. c. dorsalis* (NT, Australia) and 55 selected dipteran species (GenBank; www.ncbi.nlm.nih.gov (accessed on 10 November 2023)) were investigated using phylogenetic analyses (Appendix A). The combined mt gene set (13 PCGs) and 2 ribosomal RNAs (rRNAs) were used to construct a phylogenetic tree, with the nucleotide sequences (13 PCGs + 2 rRNAs) aligned individually using MAFFT v.7.450 [29]. The buffalo fly, *Haematobia irritans irritans* (Muscidae), was used as an outgroup for the analysis. Previously, *H. i. irritans* was used as an ideal candidate for rooting phylogenetic trees within the order Diptera because it diverged from other Dipterans relatively early in evolutionary history [8,9]. Geneious Prime v.2019.2.3 [26] was used to concatenate all 13 PCGs and 2 rRNAs after alignment. Based on the Bayesian Information Criterion (BIC), PartitionFinder v 2.1.1 [30] was used to determine the best-fit partitioning schemes and substitution models. The program MrBayes v.3.2.6 [31] was used to compute posterior probabilities (pp) using the GTR + I + G model. We ran four incrementally heated Markov chain Monte Carlo (MCMC) runs for 10,000,000 generations. The Markov chains were sampled every 200th generation, resulting in 50,000 sampled trees from each run. The initial 12,500 (25%) trees were disregarded as ‘burn-in’ (the part of the chain that was sampled prior to reaching a state of stationarity). Inferences were then drawn from the remaining 37,500 sampled trees per chain. The topologies were used to construct a majority rule consensus tree, conducting the maximum likelihood analysis using IQ-Tree v.1.6.12 [32], as implemented in the W-IQ-Tree web server [33], using 10,000 UFBoot iterations [34,35]. Substitution model estimation was performed using ModelFinder within W-IQ-TREE [36]. The resulting phylogenetic tree was visualized and annotated using FigTree v.1.4.4. (http://tree.bio.ed.ac.uk/software/figtree/ (accessed on 15 November 2023)).

## 3. Results

### 3.1. Mitochondrial Genome Organization and Base Composition Similar to Other Lucilia Species

The assembled consensus mt genome of *L. c. dorsalis* was 15,943 bp in length and was composed of 37 genes (i.e., 13 PCGs, 2 rRNAs [small (*rrnS*) and large (*rrnL*)], 22 tRNAs, and a control region), with a gene arrangement identical to those of previously published *Lucilia* mt genomes [8,9,37] (cf. Figure 1 and Appendix A). The mt genome size (15,943 bp) fell within the range of sizes reported in previously published mt genomes, such as those of *L. cuprina* strain DI213.5 (GenBank accession number: JX913753; 15,226 bp) and *L. cuprina* strain DI190.1 (JX913744; 15,952 bp) [9]. The base composition of the mt sequences was biased, with an average AT content of 77.5% (A: 39.4%, G: 9.3%, C: 12.9%, T: 38.1%), an average AT skew of 0.016, and a GC skew of −0.162. 

### 3.2. Protein-Coding Genes (PCGs) Are AT-Biased, and Codon Usage Is Dominant among Serine and Leucine Amino Acids

The total length of the PCGs was 11,123 bp, with the overall A, C, G, and T contents of the 13 PCGs being 38.2%, 13.8%, 10.2%, and 37.4%, respectively, showing a clear AT preference (Figure 2). The overall A + T and G + C contents of the protein-coding genes (PCGs) were 75.6% and 24%, respectively, with a positive AT skew (0.495) and negative GC skew (0.072) (cf. Figure 2). Incomplete stop codons were identified in the *cox2* and *nad5* genes (Appendix A). The termination codon TAA was the most frequently observed, while the initiation codons displayed the AT-rich composition. The relative synonymous codon usage (RSCU) analysis revealed a total of 33 RSCU values greater than 1.0. Serine (Ser, S) and leucine (Leu, L) were favored in codon usage, collectively accounting for 21.1% of the total 246 codons (Figure 3).

### 3.3. Transfer RNAs and Ribosomal RNAs Are AT-Rich

The *rrnS* gene was 787 bp in length and located between *trnV*(tac) and *trnI*(gat), whereas the *rrnL* gene was 1293 bp in length and located between *trnL1*(tag) and *trnV*(tac) (cf. Figure 1). The base composition of the two rRNAs was 40.1% A, 12.9% C, 7.4% G, and 39.2% T, with the AT and GC contents of the two rRNAs being 79.3% and 20.3%, respectively. The AT skew (0.011) was positive, and the GC skew (−0.270) was negative. 

The 22 tRNA genes ranged from 63 bp (*trnR*(tcg)) to 72 bp (*trnV*(tac)) in size (Figure 1). The base composition of all tRNAs was 38.8% A, 12.8% C, 10.2% G, and 38.1% T, with the AT content of the 22 tRNAs being 76.9% and with a positive AT skew (0.009) and negative GC skew (−0.113). 

### 3.4. Nucleotide Polymorphisms Were Detected within Protein-Coding Genes (PCGs)

Nucleotide polymorphisms in *L. c. dorsalis* (NT, Australia) were compared to the *L. c. dorsalis* mt reference genome from Victoria (GenBank accession number: MW255536). The *L. c. dorsalis* (NT, Australia) mt genome exhibited 28 SNPs. Among these, the majority were located in the *nad5* gene (*n* = 27), followed by the *nad2* (*n* = 9), *cox1* (*n* = 7), and *nad1* (*n* = 5) genes, respectively (Appendix A).

### 3.5. Phylogenetic Analyses Support Existing Dipteran Clades

Using a dataset of 56 mt genomes, the phylogenetic trees constructed separately using Bayesian inference (BI) and maximum likelihood (ML) methods had similar topologies, with slightly different nodal support values. As both the trees exhibited similar topologies, we integrated the support values from the ML tree into the tree obtained from BI analysis. Both trees indicated that *L. c. dorsalis* fell within the existing diversity of the genus *Lucilia* (i.e., *L. caesar*, *L. coeruleiviridis*, *L. c. cuprina*, *L. hainanensis*, *L. illustris*, *L. papuensis*, *L. porphyrina*, *L. sericata*, and *L. shenyangensis*) and had the closest relationship to *L. sericata* strains (Figure 4). The *L. c. dorsalis* mt genomes from the NT, Australia grouped with *L. cuprina* sequenced from other locations in Australia [VIC (JX913744–JX913746, MW255536), QLD (JX913749), NSW (MW255537), WA (MW255539)] and from Brazil (KT272779). Most *L. c. cuprina* collected from QLD, Australia (JX913750–X913753 and MW255538) formed a sister grouping to the *L. sericata* clade (KT272854, JX913754–JX913757, MW255540, and AJ422212), except for one QLD strain, D1213.1. The *L. cuprina* and *L. sericata* formed a sister clade to *L. caesar*, *L. hainanensis*, *L. illustris*, *L. papuensis*, *L. porphyrina*, and *L. shenyangensis*, with *L. coeruleiviridis* apparently sister to other *Lucilia* species. The members of the families Calliphoridae (labeled as clade B), Tachinidae (clade D), and Sarcophagidae (clade E) formed monophyletic groups (Figure 4). 

## 4. Discussion and Conclusions

Here, we report a consensus mitochondrial (mt) genome for *L. c. dorsalis* from the Northern Territory (NT) in Australia. The mt genome arrangement demonstrates a significant level of similarity across a diverse array of insects, indicating relative conservation for related taxa within the order Diptera [38]. Typically, the mt genomes of insects contain closed-circular and double-stranded DNA, containing 13 protein-coding genes (PCGs), 22 transfer RNA genes (tRNAs), two ribosomal RNA genes (rRNAs), and a control region [6,7]. For *L. c. dorsalis* (NT), the mt genome size (15,943 bp) was comparable to those of previously sequenced *Lucilia* species (Appendix A) [8,9,37,39], and the gene arrangement matched that of published *Lucilia* species and bore similarity to the first described mt genome of the fruit fly—*Drosophila yakuba* [40]. 

The overall nucleotide composition within the mt genome was heavily AT-biased, which accounted for 77.5% of the total mt genome. This AT richness is commonly observed for species within the family of Calliphoridae [16,41,42,43,44], which includes blowflies [8,43,45,46]. An AT bias may be attributed to the trade-off in energy efficiency, since the synthesis of A and T nucleotides requires less energy and nitrogen, compared with the nucleotides G and C [47]. The mt genome of *L. c. dorsalis* exhibited a positive average AT skew of 0.016 and a negative GC skew of −0.162, similar to the bias reported previously in the Calliphoridae family: *Chrysomya chloropyga* (AF352790, AT skew: 0.020; GC skew: −0.170) [41], *Cochliomyia hominivorax* (AF260826, AT skew: 0.034; GC skew: −0.207) [48], *L. c. cuprina* (MW255538, AT skew: 0.015; GC skew: −0.166) [8], *L. sericata* (MW255540, AT skew: 0.015; GC skew: −0.169) [8], and *L. c. dorsalis* (MW255537, AT skew: 0.016; GC skew: −0.165) [8]. These averages suggest a bias against the usage of G—a characteristic commonly observed in metazoan mt genomes [49]. The AT bias was evident in the relative codon usage of the PCGs as well [50]. Codons ending with A or T were notably more frequently utilized, contributing to the higher A + T content. 

The mt genome of *L. c. dorsalis* (NT, Australia) consisted of 28 molecular markers. Most of these markers were observed within the *nad2*, *cox2*, and *nad1* genes. These genes have historically served as species identifiers within the Calliphoridae family [20,39,51,52]. In terms of gene content in the *L. c. dorsalis* mt genome, the *cox1* gene started with a non-canonical start codon TCG (serine) [48,53]. Numerous insects do not possess the typical (ATN) start codons at the onset of the *cox1* gene, prompting the exploration of alternative (non-canonical) start codons for this gene [54]. Incomplete stop codons were detected within the *cox2* and *nad5* genes, aligning with previous observations in members of the Calliphoridae [9,48]. It is assumed that the termination codon is completed by polyadenylation [37,48]. The lengths of the *rrnL* (1293 bp) and *rrnS* (787 bp) genes were consistent with the lengths of previously sequenced *Lucilia* species [8,9]; however, establishing the boundaries of rRNA genes is challenging, due to their variability in sequence length and the absence of distinctive features [55]. The size of the 22 tRNA-encoding genes ranged from 63 bp (*trnR*(tcg)) to 72 bp (*trnV*(tac)), falling within the range observed in previously published tRNA genes of the *Lucilia* species [8,9]. 

The phylogenetic analyses provided robust evidence of relationships at both the species and sub-species levels, indicating that *L. c. cuprina* shared a closer relationship with *L. sericata*, while *L. c. dorsalis* segregated into distinct species/sub-species groupings consistent with findings in prior research [8,9,43]. The mt genome of *L. c. dorsalis* (NT, Australia; GenBank accession PP297113) grouped with those of other *L. cuprina* flies collected from different locations around Australia (MW255536, MW255537, MW255539, and JX913744 to JX913749) and Brazil (KT272779). Although *L. c. dorsalis* is the primary cause of flystrike where sheep are found in Australia [9,56], their significance in the absence of sheep needs to be further investigated. In the Northern Territory, sheep are prohibited animals, due to the presence of the bluetongue virus and its potential impact on the livestock industry [13]. However, sheep are still permitted to be moved around, into, out of, and through the NT [57], which could allow enough sheep to maintain the population of *L. c. dorsalis*. This blowfly is also a primary colonizer of fresh carrion [5], has been documented as an efficient pollinator [7], and survives as a facultative parasite in other regions of the world [58].

The mt genomes that clustered together to form the *L. sericata* clade included specimens originating from different countries, including Africa [9], Australia [8,9], UK [9], and the USA [9,59] (cf. Appendix A). In Europe, *L. sericata* is known to cause primary flystrike [16,60,61,62], whereas in Australia, this species typically plays a secondary role in flystrike [63]. Additionally, *L. cuprina cuprina* (JX913750–JX913753 and MW25538) from QLD, Australia formed a sister clade to *L. sericata*. These flies are synanthropic in behavior and are mostly prevalent in urban areas [9,51]. Reports suggest that *L. c. cuprina* is a hybrid of *L. c. dorsalis* and *L. sericata* [10,12,43]. Initially identified in Hawaii [12,43,64], *L. c. cuprina* has since been documented in Australia [9], North America [39], South Africa [52], and Southeast Asia [65]. The phylogenetic relationships inferred here within the family Calliphoridae using mt datasets support those reported in previous investigations [8,9,66].

The present study describes the mt genome of *L. c. dorsalis* from the NT, Australia and elucidates its relationship with other *Lucilia* species/subspecies and 55 dipteran taxa. Consistent with previous analyses of mt genomes of members of the Calliphoridae, the mt genome of *L. c. dorsalis* (NT, Australia) has highly conserved gene size, gene content, gene organization, and nucleotide composition. This study provides additional genetic information for further evolutionary relationship studies on blowflies within Australia and globally. Future studies will include the integration and comparison of the nuclear genomic datasets of *L. c. dorsalis* populations sourced from the NT, Australia and various regions across the country. This comprehensive analysis will provide important data to aid in further understanding their involvement in flystrike occurrences and any genetic differences between blowflies from sheep and non-sheep regions. This approach holds the potential to offer valuable insights into intricate evolutionary questions, such as cross-species hybridization and introgression.

## Figures and Tables

**Figure 1 genes-15-00506-f001:**
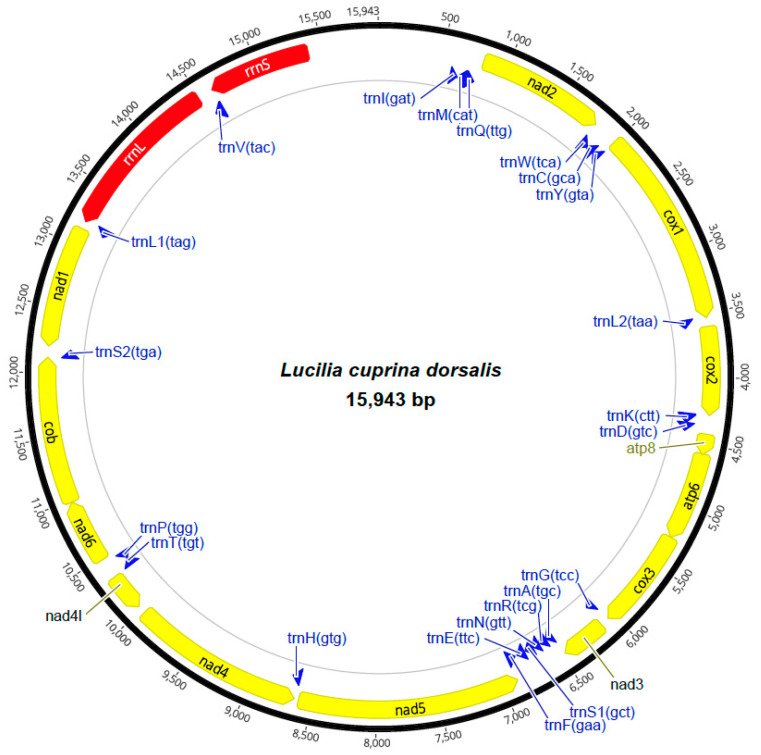
Circular representation of the mitochondrial (mt) genome of *Lucilia cuprina dorsalis* collected from the Northern Territory, Australia. Large yellow and red arrows with annotated labels situated in the mt genome map indicate the position of protein-coding genes (PCGs) and ribosomal RNA (rRNA) genes. Blue arrows with annotated labels demarcate the positions of transfer RNA (tRNA) genes. The *cox* genes refer to the cytochrome *c* oxidase subunits, *nad* genes refer to NADH dehydrogenase components, the *cob* gene refers to the cytochrome *b* gene, and *rrnL* and *rrnS* refer to ribosomal RNA genes, respectively (cf. Appendix A).

**Figure 2 genes-15-00506-f002:**
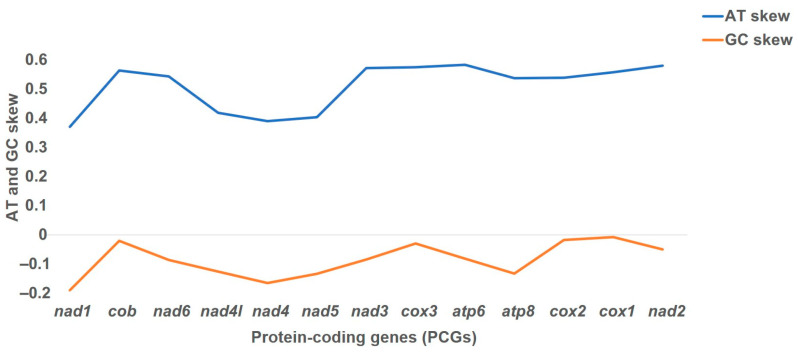
AT skew and GC skew of protein-coding genes (PCGs) in the mitochondrial genome of *Lucilia cuprina dorsalis* collected from the Northern Territory, Australia. The *x*-axis represents the protein-coding genes (PCGs), and the *y*-axis represents the AT (blue) and GC skew (orange) values associated with these PCGs.

**Figure 3 genes-15-00506-f003:**
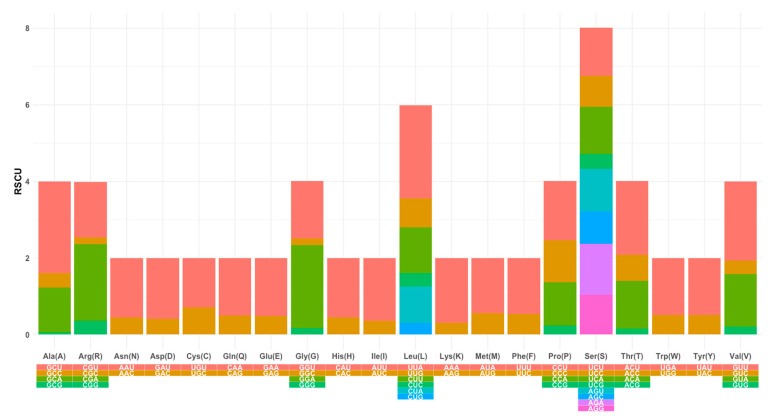
Relative synonymous codon usage (RSCU) in the protein-coding genes (PCGs) in the mitochondrial genome of *Lucilia cuprina dorsalis* collected from the Northern Territory, Australia. The different colors in the column chart represent the codon families corresponding to the amino acids listed under the columns.

**Figure 4 genes-15-00506-f004:**
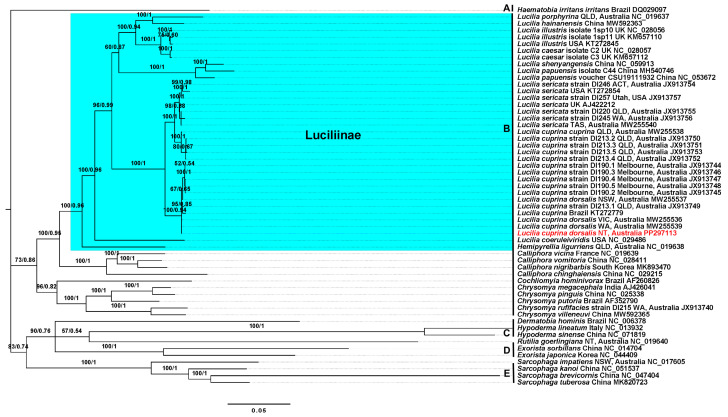
Phylogenetic relationship of *Lucilia cuprina dorsalis* (Northern Territory, Australia) with 55 members/representatives of the order Diptera. The phylogenetic tree was constructed using the Bayesian inference (BI) and maximum likelihood (ML) methods. The numbers displayed on the branches indicate bootstrap values and posterior probabilities from different analyses in the order: ML/BI. Each member is labeled with the species name, location, and GenBank accession number. *Haematobia irritans irritans* (Muscidae) was used as the outgroup. *Lucilia cuprina dorsalis* (Northern Territory, Australia) sequenced in this study is color-coded in red. The family names are labeled as A to E preceding the species names in the following order: A: Muscidae, B: Calliphoridae, C: Oestridae, D: Tachinidae, and E: Sarcophagidae. The tree branches corresponding to the subfamily Luciliinae within the Calliphoridae family are highlighted in blue. The phylogenetic tree presented here is drawn to scale, with a scale bar representing 0.05 estimated substitutions per site.

## Data Availability

The genome sequence data that support the findings of this study are openly available in GenBank of NCBI at (https://www.ncbi.nlm.nih.gov/ (accessed on 5 February 2024)) under accession no. PP297113.

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
