# Peer review of "Complete Mitochondrial Genome for Lucilia cuprina dorsalis (Diptera: Calliphoridae) from the Northern Territory, Australia"

_genes, 2024, doi:10.3390/genes15040506_

Round 1

Reviewer 1 Report

Comments and Suggestions for Authors

Authors present the complete mitochondrial genome of an Australian sheep blowfly, Lucilia cuprina dorsalis. Such genome data are available for many (sheep) blowflies from all over the world and they do hardly differ from each other. However, the here studied species is responsible for 90% of flystrike incidents in Australia and imposes an enormous economic strain on the wool industry. Moreover, the studied species is an important fertilizer of avocado and can hybridize with other Lucilia (sub)species. This makes the present results interesting for further phylogenetic studies.

The experiments were carefully done and the manuscript is professionally written. I could find only few spelling errors. 

- line 156 is not complete. See legend to Fig. 3

- line 216: either dipteran order or order Diptera

- Ref. 5: give species name in italics

- Ref. 6: use correct abbreviation for the name of the journal

- Ref. 59: give species name in italics

Author Response

We thank the editors and the three reviewers for their positive and constructive reports on our manuscript. We are pleased that all the reviewers believe that the present study represents an important and useful resource for the research community.

The present rejoinder addresses the reviewers’ suggestions and recommendations (responses in bold type) in a point-by-point manner:

Reviewer#1:

Authors present the complete mitochondrial genome of an Australian sheep blowfly, Lucilia cuprina dorsalis. Such genome data are available for many (sheep) blowflies from all over the world and they do hardly differ from each other. However, the here studied species is responsible for 90% of flystrike incidents in Australia and imposes an enormous economic strain on the wool industry. Moreover, the studied species is an important fertilizer of avocado and can hybridize with other Lucilia (sub)species. This makes the present results interesting for further phylogenetic studies.

The experiments were carefully done and the manuscript is professionally written. I could find only few spelling errors. 

- line 156 is not complete. See legend to Fig. 3

RESPONSE: We thank the reviewer for pointing this out. We have added the additional text to the following paragraph: The termination codon TAA was the most frequently observed, while the initiation codons displayed the AT-rich composition. The relative synonymous codon usage (RSCU) analysis revealed a total of 33 RSCU values greater than 1.0. Serine (Ser, S) and leucine (Leu, L) were favoured in codon usage, collectively accounting for 21.1% of the total 246 codons (Figure 3) (Line numbers: 278-283).

- line 216: either dipteran order or order Diptera

RESPONSE: We thank the reviewer for their suggestion. We have changed it to order Diptera (Line number: 354-355).

- Ref. 5: give species name in italics

RESPONSE: We thank the reviewer for pointing this out. We have italicised the species name (Line number: 498).

- Ref. 6: use correct abbreviation for the name of the journal

RESPONSE: We appreciate the reviewer for bringing these referencing errors to our attention. We have corrected the abbreviation of the name of the journal to Int. Wound J. (Line numbers: 500-501).

- Ref. 59: give species name in italics

RESPONSE: We thank the reviewer for pointing this out. We have italicised the species name (Line number: 602).

CONCLUDING REMARKS

We thank all reviewers for their time, efforts and reports, and you for the time that you have spent handling and appraising the manuscript. We have revised the manuscript to address all reviewers’ comments; the changes in the revised (R1) manuscript are highlighted in yellow. The suggested changes and recommendations have led to a much-improved manuscript, which we believe now meets the standard for publication in Genes. We look forward to the final decision on our manuscript.

 Yours sincerely,

Shilpa Kapoora,b, PhD                            Clare A. Ansteadb, PhD                                 

On behalf of all authors.

a Bio21 Molecular Science and Biotechnology Institute, The University of Melbourne, Parkville, VIC 3010, Australia

b Department of Veterinary Biosciences, Melbourne Veterinary School, Faculty of Science, The University of Melbourne | Building 400, Parkville, Victoria 3010, Australia |

shilpa.kapoor@unimelb.edu.au & clare.anstead@unimelb.edu.au

Reviewer 2 Report

Comments and Suggestions for Authors

Dear Authors,

I read carefully your submitted article Genes-2949200 and I consider it an interesting progressing on the molecular studies ,i. e. the complete mitochondrial genome of Lucilia cuprina dorsalis,  of the important group of lucilliinae in the Calliphoridae family of the order Diptera. The paper is well presented by using a clear proper technical English spelling. However, I noticed  in the review (see, please, the attached word file text) a few minor errors (loss of reference and use of not proper words). There are also 2 sentences   (94-97 lines, 120-123 lines) that need to be re-written to better understand the methods applied. The manuscript can be accepted after the minor required modifications.

Sincerely

Author Response

We thank the editors and the three reviewers for their positive and constructive reports on our manuscript. We are pleased that all the reviewers believe that the present study represents an important and useful resource for the research community.

The present rejoinder addresses the reviewers’ suggestions and recommendations (responses in bold type) in a point-by-point manner:

Reviewer #2:

 I read carefully your submitted article Genes-2949200 and I consider it an interesting progressing on the molecular studies ,i. e. the complete mitochondrial genome of Lucilia cuprina dorsalis,  of the important group of lucilliinae in the Calliphoridae family of the order Diptera. The paper is well presented by using a clear proper technical English spelling. However, I noticed  in the review (see, please, the attached word file text) a few minor errors (loss of reference and use of not proper words). There are also 2 sentences   (94-97 lines, 120-123 lines) that need to be re-written to better understand the methods applied. The manuscript can be accepted after the minor required modifications.

RESPONSE: We thank the reviewer for their helpful suggestions. We have addressed all comments in the following points:

Line number 80: “Flies” changed to “Adult flies”.

 Line number 86: Sentence modified regarding reference for Illumina: NEBNext® Ultra™ II DNA Library Prep Kit and paired-end sequenced using 2 × 150 cycles (Illumina NovaSeq 6000 platform) (Line number: 88).

 Line numbers 94-97: The sentence was rephrased to:
The mtDNA sequence was analysed using the MITOS2 [24] web server, utilizing NCBI's Invertebrate Mitochondrial translation Table 5. Additionally, transfer RNA (tRNA) genes were identified using the ARWEN software [25] (Line numbers: 111-113).

 Line numbers 120-123: The sentence was rephrased to include more details and clarification: The Bayesian phylogenetic analysis was used to compute posterior probabilities (pp) using the GTR+I+G model. We ran four incrementally heated Markov chain Monte Carlo (MCMC) runs for 10,000,000 generations. The Markov chains were sampled every 200th generation, resulting in 50,000 sampled trees from each run. The initial 12,500 (25%) trees were disregarded as ‘burn-in’ (the part of the chain that was sampled prior to reaching a state of stationarity). Inferences were then drawn from the remaining 37,500 sampled trees per chain. The topologies were used to construct a majority rule consensus tree. (Line numbers: 138-144).

We appreciate the reviewer for bringing these referencing errors to our attention. We have italicised the species names in the references (Reference numbers: 5, 6 and 59).

CONCLUDING REMARKS

We thank all reviewers for their time, efforts and reports, and you for the time that you have spent handling and appraising the manuscript. We have revised the manuscript to address all reviewers’ comments; the changes in the revised (R1) manuscript are highlighted in yellow. The suggested changes and recommendations have led to a much-improved manuscript, which we believe now meets the standard for publication in Genes. We look forward to the final decision on our manuscript.                                                                

Yours sincerely,

Shilpa Kapoora,b, PhD                            Clare A. Ansteadb, PhD                                 

On behalf of all authors.

a Bio21 Molecular Science and Biotechnology Institute, The University of Melbourne, Parkville, VIC 3010, Australia

b Department of Veterinary Biosciences, Melbourne Veterinary School, Faculty of Science, The University of Melbourne | Building 400, Parkville, Victoria 3010, Australia |

shilpa.kapoor@unimelb.edu.au & clare.anstead@unimelb.edu.au

Reviewer 3 Report

Comments and Suggestions for Authors

This manuscript describes the first mitogenome of an economically important sheep blowfly in an area where sheep are not allowed. The topic is scientifically interesting, and sufficiently novel to warrant publication. The methods are consistent with current standards and the results are well described and discussed in adequate detail. All in all, this is a well written and valuable manuscript.

I have only a few minor edits and suggestions:

Line 22: … the same as that of previously …

Line 23: genomes = mitogenomes

Line 46: replace “(Wiedemann)” with “(Wiedemann, 1830)”

Line 111: The phylogenetic relationships between …

Line 116: add a brief statement (and reference) why Haematobia irritans is an appropriate outgroup

Line 156: delete the dash before “Codon usage”

Lines 163 and201 : Figures 3 and 4 have rather low resolution, and most of the text is poorly legible. Please make sure the final version has much higher resolution.

Line 169: while = whereas

Line 185: Using a dataset of 56 mitogenomes, …

Line 197: ancestral = sister  

Line 199: clades are always monophyletic, So please either use “monophyletic groups” or “clades”

Line 201: The tree shows support values from Bayesian and ML analyses, but the tree itself can only have been made using one of the two methods, so please clarify which method was used.

Lines 231-234: please use italics only for species names.

Lines 259-267: These statements do not refer to phylogenetic results, and are best transferred to a new paragraph after line 278.

Line 264: the word “relationship” may be deleted

Author Response

We thank the editors and the three reviewers for their positive and constructive reports on our manuscript. We are pleased that all the reviewers believe that the present study represents an important and useful resource for the research community.

The present rejoinder addresses the reviewers’ suggestions and recommendations (responses in bold type) in a point-by-point manner:

Reviewer #3: This manuscript describes the first mitogenome of an economically important sheep blowfly in an area where sheep are not allowed. The topic is scientifically interesting, and sufficiently novel to warrant publication. The methods are consistent with current standards and the results are well described and discussed in adequate detail. All in all, this is a well written and valuable manuscript.

I have only a few minor edits and suggestions:

Line 22: … the same as that of previously …

RESPONSE: We thank the reviewer for their suggestion. We have changed the sentence to “The gene order of the current mt genome is consistent with the previously published L. cuprina mt genomes” (Line number: 22-23).

Line 23: genomes = mitogenomes

RESPONSE: We have changed it to mt genomes (Line number: 23)

Line 46: replace “(Wiedemann)” with “(Wiedemann, 1830)”

RESPONSE: We thank the reviewer for pointing this out. We have changed it to (Wiedemann, 1830) (Line number: 48).

Line 111: The phylogenetic relationships between …

RESPONSE: We changed the sentence to “The phylogenetic relationships between L. c. dorsalis (NT, Australia) and 55 selected dipteran species available on GenBank (www.ncbi.nlm.nih.gov) was investigated using phylogenetic analyses (Table S1).” (Line number: 127).

Line 116: add a brief statement (and reference) why Haematobia irritans is an appropriate outgroup

RESPONSE: We express our gratitude to the reviewer for recommending the inclusion of a brief statement and reference about Haematobia irritans. We have added the explanation as recommended: “Previously, H. i. irritans has been used as an ideal candidate for rooting phylogenetic trees within the order Diptera because it diverged from other dipterans relatively early in evolutionary history.” (Line numbers: 132-135).

Line 156: delete the dash before “Codon usage”

RESPONSE: We have deleted the dash before Codon usage (Line numbers: 278-283).

Lines 163 and 201: Figures 3 and 4 have rather low resolution, and most of the text is poorly legible. Please make sure the final version has much higher resolution.

RESPONSE: We thank the reviewer for their suggestion. We have also provided the high-resolution figures which will be used for the final version of the manuscript.

Line 169: while = whereas
RESPONSE: We thank the reviewer for the suggestion. We have changed to “whereas” (line number: 296).

Line 185: Using a dataset of 56 mitogenomes, …

RESPONSE: We have changed the sentence to “Using a dataset of 56 mt genomes the phylogenetic trees constructed separately using Bayesian inference (BI) and maximum likelihood (ML) methods had similar topologies with slightly different nodal support values (Line number: 318).

Line 197: ancestral = sister  

RESPONSE: We have changed the word ancestral to “sister” (Line number: 332).

Line 199: clades are always monophyletic, So please either use “monophyletic groups” or “clades”

RESPONSE: We thank the reviewer for the suggestion. We have changed the monophyletic clades to “monophyletic groups” (Line number: 334).

Line 201: The tree shows support values from Bayesian and ML analyses, but the tree itself can only have been made using one of the two methods, so please clarify which method was used.

RESPONSE: We thank the reviewer for pointing this out. The tree was constructed using the Bayesian and ML methods. As both trees exhibited similar topologies, we integrated the support values from the ML tree into the tree obtained from Bayesian analysis. We have incorporated the clarification in the manuscript (Line numbers: 320-321).

Lines 231-234: please use italics only for species names.

RESPONSE: We thank the reviewer for pointing this out. We have only used the italics for species names (Line number: 367-399).

Lines 259-267: These statements do not refer to phylogenetic results and are best transferred to a new paragraph after line 278.

RESPONSE: We thank the reviewer for suggesting the change. We would like to keep these statements together in the discussion section explaining the phylogenetic results of L. c. dorsalis clade. We want to make a point/emphasize that this clade consists of both the L. c. dorsalis (parasitic and non-parasitic) collected from different parts of Australia (Line number: 424-432).

Line 264: the word “relationship” may be deleted

RESPONSE: We could not find the word relationship in Line 264.

CONCLUDING REMARKS

We thank all reviewers for their time, efforts and reports, and you for the time that you have spent handling and appraising the manuscript. We have revised the manuscript to address all reviewers’ comments; the changes in the revised (R1) manuscript are highlighted in yellow. The suggested changes and recommendations have led to a much-improved manuscript, which we believe now meets the standard for publication in Genes. We look forward to the final decision on our manuscript.

Yours sincerely,

Shilpa Kapoora,b, PhD                             Clare A. Ansteadb, PhD                                 

On behalf of all authors.

a Bio21 Molecular Science and Biotechnology Institute, The University of Melbourne, Parkville, VIC 3010, Australia

b Department of Veterinary Biosciences, Melbourne Veterinary School, Faculty of Science, The University of Melbourne | Building 400, Parkville, Victoria 3010, Australia |

shilpa.kapoor@unimelb.edu.au & clare.anstead@unimelb.edu.au